# Disruption of the NF-κB/IL-8 Signaling Axis by Sulconazole Inhibits Human Breast Cancer Stem Cell Formation

**DOI:** 10.3390/cells8091007

**Published:** 2019-08-30

**Authors:** Hack Sun Choi, Ji-Hyang Kim, Su-Lim Kim, Dong-Sun Lee

**Affiliations:** 1School of Biomaterials Sciences and Technology, College of Applied Life Science, Jeju National University, Jeju 63243, Korea; 2Subtropical/tropical Organism Gene Bank, Jeju National University, Jeju 63243, Korea; 3Interdisciplinary Graduate Program in Advanced Convergence Technology & Science, Jeju National University, Jeju 63243, Korea

**Keywords:** sulconazole, NF-κB, IL-8, mammosphere, breast cancer stem cells

## Abstract

Breast cancer stem cells (BCSCs) are tumor-initiating cells that possess the capacity for self-renewal. Cancer stem cells (CSCs) are responsible for poor outcomes caused by therapeutic resistance. In our study, we found that sulconazole—an antifungal medicine in the imidazole class—inhibited cell proliferation, tumor growth, and CSC formation. This compound also reduced the frequency of cells expressing CSC markers (CD44^high^/CD24^low^) as well as the expression of another CSC marker, aldehyde dehydrogenase (ALDH), and other self-renewal-related genes. Sulconazole inhibited mammosphere formation, reduced the protein level of nuclear NF-κB, and reduced extracellular IL-8 levels in mammospheres. Knocking down NF-κB expression using a p65-specific siRNA reduced CSC formation and secreted IL-8 levels in mammospheres. Sulconazole reduced nuclear NF-κB protein levels and secreted IL-8 levels in mammospheres. These new findings show that sulconazole blocks the NF-κB/IL-8 signaling pathway and CSC formation. NF-κB/IL-8 signaling is important for CSC formation and may be an important therapeutic target for BCSC treatment.

## 1. Introduction

Breast cancer is a cancer that develops from common breast tissue and a major fatal health problem among females [1]. Patients treated with different therapeutics suffer from cancer relapse and metastasis because of cancer stem cells (CSCs), a subpopulation of tumor cells. CSCs are heterogeneous bulk tumor cells that differentiate into cancer cells. CSCs are resistant to chemotherapies and contribute to tumor heterogeneity [2]. CSCs were first identified in leukemia and found to show properties similar to those of stem cells by Bonnet and Dick [3]. Markers of breast cancer stem cells (BCSCs) include CD44, CD133, and ALDH1. CD44 expression is upregulated in the microenvironment that promotes cancer progression and metastasis [4]. Additionally, CD44 isoforms are reliable markers of CSCs. The CD44 isoform CD44v-xCT regulates redox in cancer stem cells [5]. The signaling pathways regulating CSC stemness and differentiation are the Wnt, Hedgehog, Hippo, and Notch signaling pathways. Molecular targeting of these pathways to inhibit BCSCs may be a useful tool for cancer treatment [6]. Sox2, Nanog, Oct4, and c-Myc are crucial for CSC formation and potential targets for cancer therapy. One report showed that NF-κB was involved in CSCs from primary acute myeloid leukemia (AML) samples [7]. Additionally, BCSCs overexpress NF-κB signaling pathway components and induce NF-κB activity. BCSCs have high protein expression of NF-κB [8]. Inhibiting NF-κB signaling with BMS-345541 in lung cancer reduces the stemness and self-renewal capacity of lung CSCs [9]. The cytokines IL-6 and IL-8 regulate links between CSCs and the microenvironment. The Stat3 and NF-κB pathways regulate the gene expression of IL-6 and IL-8 in breast cancer. Microenvironmental IL-6 and IL-8 regulate BCSC populations [10]. In lung cancer patients, high extracellular IL-6 levels are associated with a poor prognosis [11,12]. IL-6 regulates BCSC formation through the IL-6/IL-6 receptor interaction [13]. The protein expression level of IL-8 is higher in breast cancer cells than in normal breast tissue cells, and IL-8 promotes cancer progression. IL-8 promotes BCSC activity through the CXCR1/IL8 interaction. IL-8/CXCR1 signaling is an important pathway for targeting BCSCs [14]. Azole compounds used as antifungal drugs inhibit the ergosterol biosynthesis pathway through suppression of the enzyme lanosterol 14-α-demethylase, a cytochrome P450 (CYP) enzyme. Azole antifungal drugs consist of an imidazole (clotrimazole and ketoconazole) and a triazole (fluconazole and itraconazole). Recently, antifungal imidazole drugs have well-established pharmacokinetic profiles and known toxicity, which can make these generic drugs strong candidates for repositioning as antitumor therapies [15]. Sulconazole is an antifungal medicine in the imidazole class and has broad-spectrum activity against dermatophytes [16]. We demonstrated that sulconazole had antiproliferative properties in breast cancer and inhibited BCSC formation through a reduction in IL-8 expression induced by disrupting the NF-κB pathway.

## 2. Materials and Methods

### 2.1. Cell Lines and Media

MCF-7 and MDA-MB-231 cells were grown in Dulbecco’s modified Eagle’s medium (DMEM; Gibco, Thermo Fisher Scientific, Waltham, CA, USA) supplemented with 10% (v/v) fetal bovine serum (FBS; Thermo Fisher Scientific, Waltham, CA, USA), 1% penicillin and streptomycin in a humidified 5% CO_2_ incubator at 37 °C. Breast cancer cells were cultured at a concentration of 3.5 × 10^4^ or 0.5 × 10^4^ cells/well in an Ultralow Adherent plate containing MammoCult^TM^ medium (STEMCELL Technologies, Vancouver, BC, Canada) supplemented with heparin and hydrocortisone in a humidified 5% CO_2_ incubator at 37 °C. A 6-well plate was scanned, and mammosphere counting was performed using the NICE program [17]. A mammosphere formation assay was determined by evaluating mammosphere formation efficiency (MFE) (%) as previously described [18].

### 2.2. Antibodies, siRNAs, and Plasmids

Anti-pStat3 (Y705) (rabbit monoclonal) antibodies were obtained from Cell Signaling Technology. Anti-p65 (mouse polyclonal), anti-pp65, anti-Stat3 (rabbit monoclonal), anti-β-actin (mouse polyclonal), and anti-Lamin b antibodies were purchased from Santa Cruz Biotechnology. Anti-CD44 FITC-conjugated and anti-CD24 PE-conjugated antibodies were obtained from BD Pharmingen. A human p65-specific siRNA and scrambled siRNA were purchased from Bioneer (Daejeon, Korea).

### 2.3. Cell Proliferation

We used a previously reported method [19]. Breast cancer cells were incubated in a 96-well plate with sulconazole for 24 h. We followed the manufacturer’s protocol for a CellTiter 96^®^ Aqueous One Solution cell kit (Promega), and the optical density at 490 nm (OD_490_) was determined using a plate reader (SpectraMax, Molecular Devices, San Jose, CA, USA).

### 2.4. Colony Formation and Migration Assays

MDA-MB-231 cells were cultured at 2 × 10^3^ cells/well with different concentrations of sulconazole in DMEM/10% FBS. The cancer cells were incubated, and colonies were counted. The cancer cells were incubated in a 6-well plate, and a scratch was made using a microtip. After washing with DMEM, the breast cancer cells were cultured with sulconazole. We followed a previously described method [20].

### 2.5. Flow Cytometry Analysis of the Expression of CD24 and CD44 and an ALDEFLUOR Assay

We used a previously described method [20]. In total, 1 × 10^6^ cells were incubated with FITC-conjugated anti-CD44 and PE-conjugated anti-CD24 antibodies (BD, San Jose, CA, USA) and incubated on ice for 20 min. The breast cancer cells were washed two times with 1X PBS and assayed by using a flow cytometer (BD curi C6, San Jose, CA, USA). An ALDH1 assay was performed using an ALDEFUOR kit (STEMCELL Technologies, Vancouver, BC, Canada). We followed a previously described method [20]. Breast cancer cells were incubated in ALDH assay buffer at 37 °C for 20 min. ALDH-positive cells were determined by using a personal flow cytometer (BD Accuri C6).

### 2.6. RNA Isolation and Real-Time RT-qPCR

Total RNA was purified, and RT-qPCR was performed using a one-step RT-qPCR kit (Takara, Tokyo, Japan). We followed a previously described method [19]. The specific primers used can be found in Appendix A. The β-actin gene was used as an internal control for RT-qPCR.

### 2.7. Immunoblot Analysis

Proteins isolated from breast cancer cells and mammospheres were separated by 10% SDS-PAGE and transferred to a polyvinylidene fluoride (PVDF) membrane (EMD Millipore, Burlington, MA, USA). The blots were blocked in 5% skim milk in 1X PBS-Tween 20 at room temperature for 60 min and then incubated overnight at 4 °C with primary antibodies. The antibodies were anti-JAK2, anti-Stat3, anti-p65, anti-pp65, anti-lamin B, anti-phospho-Stat3 (Cell Signaling, Danvers, MA, USA), and anti-β-actin (Santa Cruz Biotechnology, Dallas, TA, USA) antibodies. After washing, the blots were detected with IRDye 680 RD and 800 CW secondary antibodies, and images were detected by using ODYSSEY CLx (LI-COR, Lincoln, NE, USA).

### 2.8. Electrophoretic Mobility Shift Assays (EMSAs)

Nuclear extracts were prepared as described previously [21]. An EMSA for NF-κB binding was performed using an IRDye 800-labeled NF-κB consequence oligonucleotide (LI-COR) for 30 min at room temperature. Samples were run on a nondenaturing 6% PAGE gel, and EMSA data were captured by ODYSSEY CLx (LI-COR). Supershifts were analyzed by incubating nuclear extracts for 30 min before the addition of the IRDye 800-labeled NF-κB consequence oligonucleotide.

### 2.9. In Vivo Mouse Experiments

Twelve female BALB/C nude mice were injected with MDA-MB-231 cells and treated with/without sulconazole (10 mg/kg). Tumor volume was measured after 1.5 months using a formula (Figure 2). Mouse experiments were performed as described previously [22]. Animal care and animal experiments were conducted in accordance with protocols approved by the Jeju National University Animal Care and Use Committee. Female BALB/C nude mice (5 weeks old) were obtained from OrientBio (Seoul, Korea) and kept in mouse facilities for 7 days. Twelve female BALB/C nude mice injected with MDA-MB-231 cells were monitored. Nude mice (*n* = 6) received sulconazole using mammary fat pad injection with an optimized dosage of 10 mg/kg. The dose of drug used was 10 mg/kg (200 μg/100 μL) once a week. The measurement was made every 3 to 4 days starting from day 10. The solvent used is DMSO. Tumor volumes were measured using the formula: V = (width^2^ × length)/2.

### 2.10. Statistical Analysis

All data from three independent experiments are shown as the mean ± standard deviation (SD). Data were analyzed using one-way ANOVA. A *p*-value less than 0.05 was considered statistically significant.

## 3. Results

### 3.1. Sulconazole Inhibits the Proliferation of Breast Cancer Cells

We examined the antiproliferative effect of sulconazole on human breast cancer cells. Sulconazole inhibited proliferation (Figure 1A,B). Apoptosis in breast cancer cells was induced by sulconazole at a concentration of 20 μM (Figure 1C). Sulconazole induced caspase3/7 activity in breast cancer cells (Figure 1D). The breast cancer cells showed formation of apoptotic bodies in response to sulconazole treatment (Figure 1E). Sulconazole inhibited the migration of cancer cells and reduced the number of colonies (Figure 1F,G). Our data showed that sulconazole effectively inhibited proliferation, migration, and colony formation.

### 3.2. Sulconazole Inhibits Tumor Growth

As sulconazole has cytotoxic activity in breast cancer, we tested whether sulconazole inhibits tumor growth in an in vivo mouse model. The tumor volume in sulconazole-injected mice was smaller than that in control mice (Figure 2A). The tumor weights in the sulconazole-injected mice were lower than those in the control mice (Figure 2B). The sulconazole-treated mice showed body weights similar to those of the control mice (data not shown). Our data showed that sulconazole effectively decreased tumor growth in the xenograft mouse model.

### 3.3. Effect of Sulconazole on the Properties of BCSCs

To examine whether sulconazole inhibits mammosphere formation, we treated mammospheres derived from breast cancer cells (MCF-7 and MDA-MB-231) with different concentrations of sulconazole. Sulconazole inhibited mammosphere formation. The number of mammospheres declined by 90%, and mammosphere size also decreased (Figure 3A,B). CD44^+^/CD24^-^ cancer cells were assessed under sulconazole treatment. Sulconazole reduced the percentage of CD44^+^/CD24^-^ cells from 14.23% to 3.53% (Figure 4A). Additionally, we performed an ALDEFLUOR assay to examine the effect of sulconazole on ALDH-positive cells. Sulconazole reduced the ALDH-positive cell percentage from 3.2% to 1.5% (Figure 4B). Our data show that sulconazole inhibits BCSCs.

### 3.4. Sulconazole Inhibits Mammosphere Formation Through the Inhibition of p65 Nuclear Translocation

To understand the molecular mechanism of sulconazole in mammosphere formation, the nuclear translocation of p65 was evaluated in mammospheres. Our data showed that nuclear phosphor-p65 and p65 levels were reduced significantly in a dose-dependent manner under sulconazole treatment (Figure 5A). Because caffeic acid phenethyl ester (CAPE) inhibits the nuclear translocation of p65 and the activation of the NF-κB signaling pathway [23], we evaluated mammosphere formation after treatment with CAPE. CAPE inhibited mammosphere formation. As a result of the use of sulconazole and CAPE, we showed that the inhibition of p65 nuclear translocation blocked mammosphere formation (Figure 5C). To examine p65 function in BCSCs, we tested the effect of NF-kB using a p65-specific siRNA. siRNA-p65 inhibited mammosphere formation in breast cancer (Figure 5B). In conclusion, we show that NF-κB regulates CSC formation and a CSC survival factor (Figure 5).

### 3.5. Sulconazole Inhibits the NF-κB Signaling Pathway and Production of Extracellular IL-8 in Mammospheres

To analyze the biological function of sulconazole, we examined NF-κB signaling and the extracellular IL-8 level in mammospheres treated with sulconazole. Compared with a vehicle, sulconazole reduced nuclear p65 protein levels (Figure 6A). We checked NF-κB binding with sulconazole-treated nuclear proteins using an IRDye 800-NF-κB probe that binds an NF-κB oligonucleotide with high affinity. Sulconazole reduced the ability of p65 to bind to the IRDye 800-NF-κB probe (Figure 6B, lane 3). NF-κB/IRDye 800-NF-κB probe specificity was confirmed using a 10-fold increased concentration of self-competitor oligonucleotides (Figure 6B, lane 4). Sulconazole decreased the DNA-binding capacities of NF-κB. Extracellular IL-6 and IL-8 have essential functions in CSC formation [13]. NF-κB regulated the transcription of the IL-6 and IL-8 genes, binding to the promoter regions of the IL-6 and IL-8 genes. To assess the transcriptional levels of IL-6 and IL-8 under sulconazole treatment, we performed real-time RT-qPCR analysis of mammospheres using IL-6- and IL-8-specific primers. The transcript data showed that sulconazole reduced the transcript level of IL-8 but not that of IL-6 (Figure 6C). After using a siRNA targeting p65, the transcript data showed that sulconazole reduced the transcript level of IL-8 (Figure 6D). To test the level of extracellular IL-8, we performed cytokine profiling of the culture medium from mammospheres. After sulconazole treatment, the cytokine profiling data showed that sulconazole reduced the level of extracellular IL-8 but not that of IL-6 (Figure 6E).

### 3.6. Sulconazole Inhibits Stem Cell Marker Gene Expression and Mammosphere Growth

To determine whether sulconazole regulates stem cell marker genes, we tested the transcription of stem cell marker genes. Sulconazole inhibited the expression of genes such as Nanog, c-Myc, and CD44 in BCSCs (Figure 7A). To verify that sulconazole reduces mammosphere growth, we added sulconazole to a mammosphere culture and counted the mammosphere cells. Sulconazole induced cell death in the mammospheres. These data showed that sulconazole led to a dramatic reduction in mammosphere growth (Figure 7B). These data showed that NF-κB signaling was essential for regulating mammosphere growth and that sulconazole inhibited mammosphere formation through deregulation of the NF-κB/IL-8 signaling pathway.

## 4. Discussion

Breast cancer is a female cancer that develops in the breast tissue. Breast cancer treatment using chemotherapy and radiotherapy eradicates the primary tumor, resulting in an increased survival rate in breast cancer patients [24]. Cancer metastasis and relapse have been attributed to CSC existence after chemotherapy [25]. BCSCs remain incompletely understood and are potential targets for breast cancer therapies [26]. The minimum biomarkers for BCSCs are the cell-surface markers CD44+/CD24- and CD44 upregulation is linked to tumor formation [27].

Our data show that sulconazole has potential as an antitumor and anti-CSC agent for breast cancer therapy. Sulconazole inhibits breast cancer hallmarks (Figure 1) and BCSC hallmarks (Figure 3 and Figure 4). It is well known that the maintenance of BCSC properties is regulated by Stat3 [19,28,29]. We checked the Stat3 signaling pathway in the context of sulconazole treatment, but sulconazole did not regulate the Stat3 signaling pathway (Figure 6). The involvement of the NF-κB signaling pathway has been observed in primary AML samples, and elevated or constitutive NF-κB signaling activation is known to be present in many solid tumor types [30]. A high level of nuclear p65 is an essential feature of CSC formation [31]. As the NF-κB signaling pathway is important for BCSC survival, we examined the localization of the p65 subunit. Our results showed that sulconazole inhibited the translocation of p65. The inhibition of p65 translocation induced the inhibition of BCSC formation. CAPE is a strong specific inhibitor of NF-κB and prevents the nuclear translocation of p65 [32]. CAPE inhibited the translocation of p65 and pp65 and induced the inhibition of mammosphere formation. A siRNA specific for p65 inhibited mammosphere formation. Pyrrolidinedithiocarbamate (PDTC), another NF-κB pathway inhibitor, is known to inhibits CSC formation [33]. The imidazole-class drug BMS-345541 IKK inhibitor as an NF-kB inhibitor reduced their stem cell concentrations and self-renewal capacity in lung cancer cells. The nuclear levels of p65 and NF-κB signaling are important for BCSC survival.

Tumor progression and CSC survival can be regulated by the cytokines IL-6 and IL-8 in an autocrine or paracrine manner [10,34,35]. The JAK2/STAT3/IL-6 pathway is hyperactivated in several types of cancer and important to the growth of BCSCs [28]. This hyperactivation is related to a poor prognosis [36]. Extracellular IL-8 is overexpressed in triple-negative breast cancer (TNBC) and is an important therapeutic target in TNBC [37]. IL-8 signaling is an important key for targeting BCSCs [14]. We know that NF-κB can regulate the transcriptional regulation of the IL-6 and IL-8 genes in BCSCs. We assessed IL-6 and IL-8 gene transcripts in BCSCs treated with a p65 translocation inhibitor, sulconazole, and a siRNA specific for p65 that induced p65 downregulation. Both conditions showed that the RNA level of IL-8 was lower in treated samples than in control samples, but the RNA level of IL-6 was not changed between the treated samples and the control samples (Figure 6). Sulconazole reduced CSC formation through downregulation of NF-κB/IL-8 in breast cancer.

Sulconazole is an antifungal and antibacterial medicine in the imidazole class and is available as a cream to treat skin infection. Sulconazole inhibits the growth of common pathogenic dermatophytes by blocking sterol 14α-demethylase (CYP51) [38]. In this study, we first showed that sulconazole has cytotoxicity against breast cancer cells and reduces BCSC characteristics. These results support sulconazole as an important therapeutic agent to inhibit breast cancer and BCSCs.

## 5. Conclusions

In this study, we found that sulconazole—an antifungal medicine in the imidazole class—inhibited cell proliferation, tumor growth, and CSC formation. This compound also reduced the frequency of cells with the CSC marker phenotype of CD44^+^/CD24^−^ as well as the expression of ALDH and self-renewal-related genes. Sulconazole inhibited mammosphere formation and reduced the protein level of nuclear NF-κB. NF-κB knockdown using a p65-specific siRNA reduced CSC formation and secreted IL-8 levels in mammospheres. Sulconazole reduced nuclear NF-κB protein levels and extracellular IL-8 levels. These new findings showed that sulconazole blocked the NF-κB/IL-8 signaling pathway and CSC formation. NF-κB/IL-8 signaling is important for CSC formation and may be an important therapeutic target for BCSC treatment.

## Figures and Tables

**Figure 1 cells-08-01007-f001:**
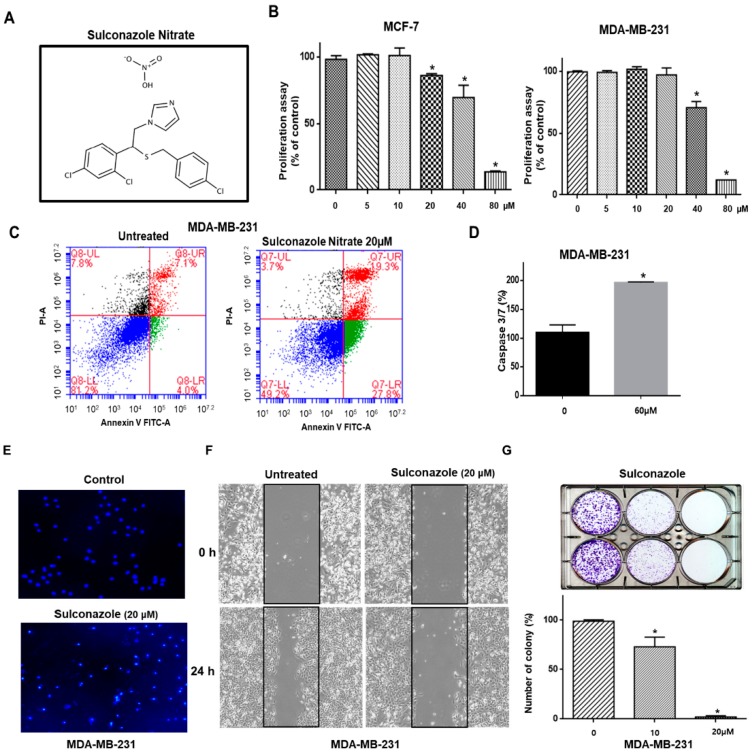
Sulconazole inhibits cell proliferation in breast cancer. (**A**) The molecular structure of sulconazole is shown. (**B**) Breast cancer cells were incubated in a 96-well plate with the indicated concentration of sulconazole. Cell proliferation was measured by an MTS assay. (**C**) Sulconazole induced apoptosis in cancer cells at the indicated concentration. Apoptotic cells were determined using Annexin V/PI staining. (**D**) The caspase3/7 activity of cancer cells was determined using a Caspase-Glo 3/7 assay kit (Promega). The data are presented as the mean ± SD; *n* = 3 independent experiments; * *p* < 0.05 vs. the control (0.3% DMSO). (**E**) Apoptotic cells were analyzed by fluorescence nuclear staining using Hoechst 33,258 dye (magnification, 40×). (**F**) The effect of sulconazole on the migration of cancer cells was evaluated using a scratch assay. The scratch assay was performed with cancer cells treated with sulconazole. (**G**) The effect of sulconazole on colony formation is shown. 1000 cancer cells were incubated in 6-well plates with sulconazole (0.1% DMSO) and 0.1% DMSO. Representative images were recorded. The data are presented as the mean ± SD; *n* = 3 independent experiments; * *p* < 0.05 vs. the control.

**Figure 2 cells-08-01007-f002:**
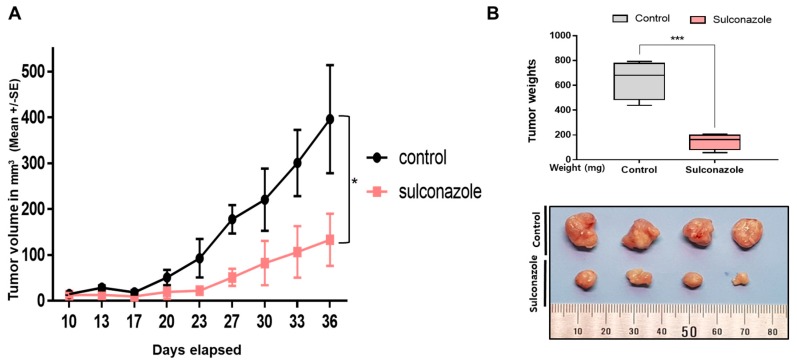
Effect of sulconazole on in vivo tumor growth. (**A**) NOD-SCID nude mice were inoculated with MDA-MB-231 cells and treated with sulconazole or vehicle. The dose of drug used was 10 mg/kg once a week. Tumor volume was measured at the indicated time points using a caliper and calculated as (width^2^ × length)/2 and are reported (Mean ± SE). (**B**) The effect of sulconazole on tumor weights was evaluated. Tumor weights were assayed after sacrifice. Photographs were taken of isolated tumors from control or sulconazole-treated mice. * *p* < 0.05 and *** *p* < 0.05 vs. the control.

**Figure 3 cells-08-01007-f003:**
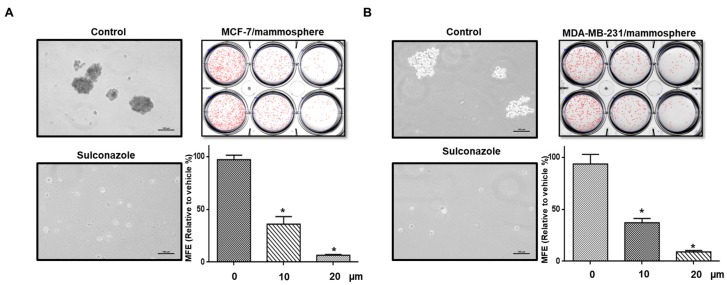
Effect of sulconazole on the mammosphere-forming ability of breast cancer cells. (**A**, **B**) Effect of sulconazole on the mammosphere formation of breast cancer cells. To establish mammospheres, MCF-7 and MDA-MB-231 cells were seeded at a density of 4 × 10^4^ and 1 × 10^4^ cells/well, respectively, in ultralow attachment 6-well plates containing 2 mL of complete MammoCult^TM^ medium (StemCell Technologies) which was supplemented with 4 μg/mL heparin, 0.48 μg/mL hydrocortisone, 100 U/mL penicillin, and 100 μg/mL streptomycin. Mammospheres were cultured with sulconazole (10 or 20 μM) solubilized in 0.05% DMSO or 0. 1% DMSO. The breast cancer cells were incubated with sulconazole in CSC culture medium for 7 days. A mammosphere formation assay evaluated mammosphere formation efficiency (MFE, % of control), which corresponds to the number of mammospheres per well/the number of total cells plated per well ×100 as previously described (scale bar = 100 μm) [22]. The data are presented as the mean ± SD; *n* = 3 independent experiments;* *p* < 0.05 vs. the control (0. 1% DMSO).

**Figure 4 cells-08-01007-f004:**
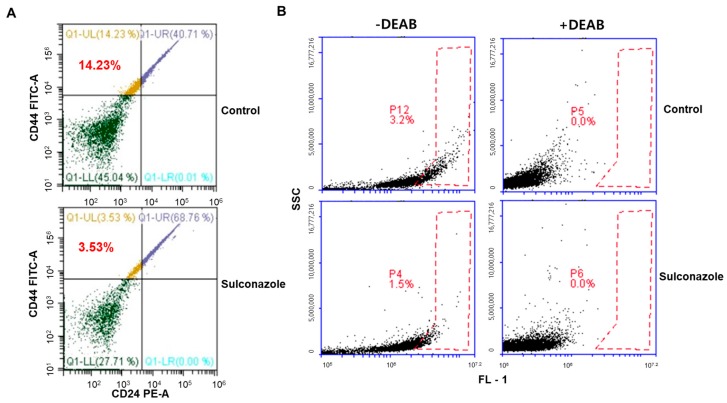
Effect of sulconazole on CD44^+^/CD24^−^-and ALDH-positive cell populations. (**A**) The CD44^+^/CD24^−^ cell population treated with sulconazole (20 μM) was assayed by flow cytometry. For FACS analysis, 10,000 cells were assayed. Gating was based on binding of the control antibody (Red cross). (**B**) ALDH-positive cells were detected by using an ALDEFLUOR kit. A representative flow cytometry dot plot is shown. The right panel indicates ALDH-positive cells treated with the ALDH inhibitor DEAB (7.5 μM), and the left panel shows ALDH-positive cells without DEAB treatment. The ALDH-positive population was gated in a box (red dot line box).

**Figure 5 cells-08-01007-f005:**
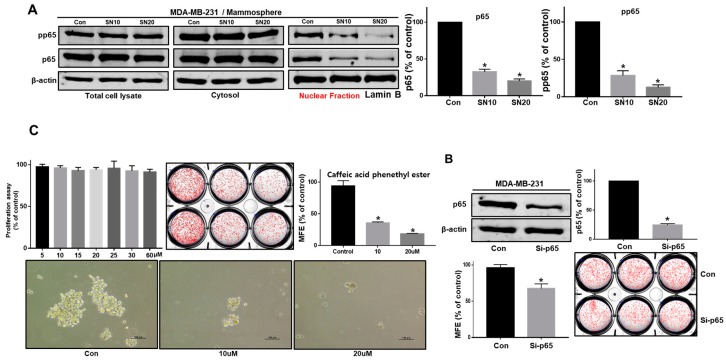
Sulconazole inhibits mammosphere formation through disruption of NF-κB activity. (**A**) Cancer cells were treated with sulconazole for 24 h. Nuclear and cytosolic proteins were run on a 10% SDS-PAGE gel, followed by immunoblotting with anti-p65 and anti-pp65 antibodies. (**B**) The effect of knocking down p65 expression using a siRNA specific for p65 on mammosphere formation was evaluated. The p65-knockdown effect was confirmed by immunoblotting using an anti-p65 antibody. (**C**) The effect of caffeic acid phenethyl ester, an NF-κB-specific inhibitor, on mammosphere formation was evaluated. A mammosphere formation assay evaluated mammosphere formation efficiency (MFE) (scale bar = 100 μm). The data are presented as the mean ± SD; *n* = 3 independent experiments;* *p* < 0.05 vs. the control.

**Figure 6 cells-08-01007-f006:**
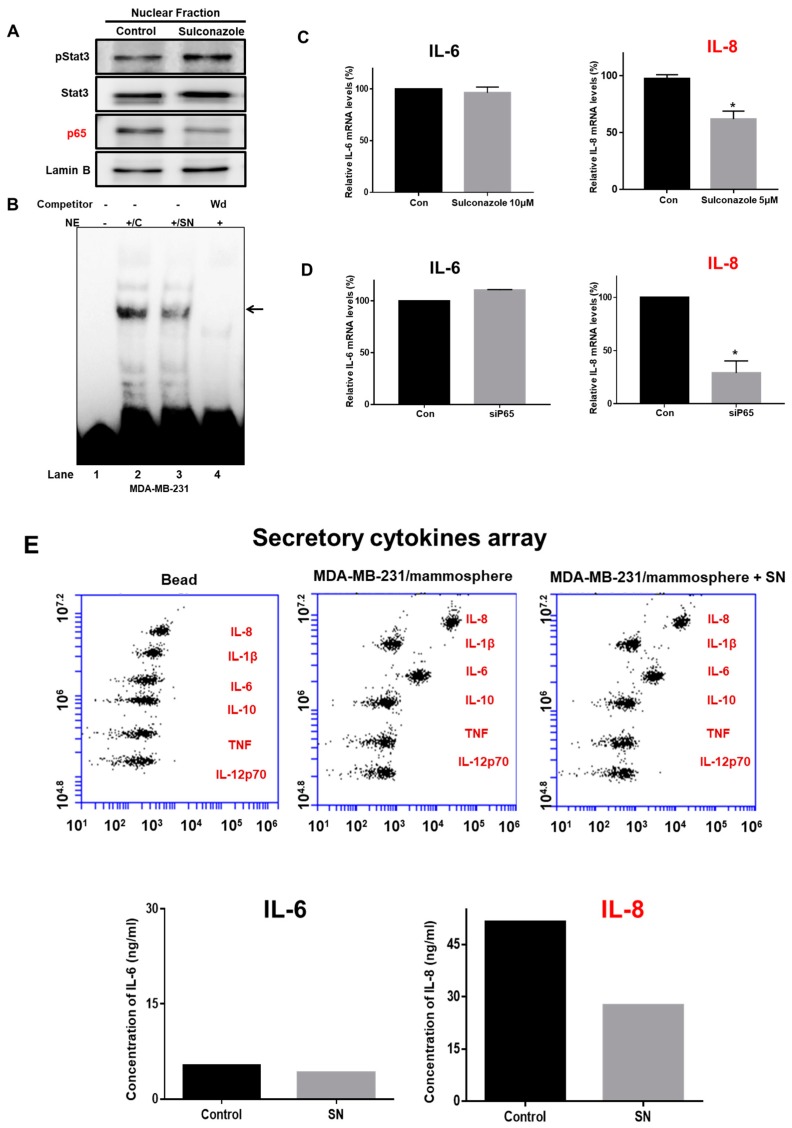
Effect of sulconazole on the NF-κB and IL-8 signaling pathways. (**A**) Cancer cells were treated with sulconazole for 24 h. Nuclear proteins were resolved on a 10% SDS-PAGE gel, followed by western blotting with anti-pStat3, anti-Stat3, anti-p65, and anti-Lamin B antibodies. (**B**) An electrophoretic mobility shift assay (EMSA) was used to assess nuclear lysates from mammospheres treated with sulconazole. The nuclear proteins were incubated with an IRDye 800-NF-κB probe and separated by 6% PAGE. Lane 1: probe only; lane 2: nuclear proteins with probe; lane 3: sulconazole-treated nuclear proteins with probe; lane 4: 10× self-competition. The arrow indicates the DNA/NF-κB interaction in the nuclear lysates. (**C**,**D**) Transcriptional levels of the IL-6 and IL-8 genes were determined in sulconazole-treated mammospheres and p65-knockout samples treated with a siRNA specific for p65. IL-6- and IL-8-specific primers were used for real-time RT-PCR. β-actin acted as an internal control. The data are presented as the mean ± SD; *n* = 3 independent experiments; * *p* < 0.05 vs. the control. (**E**) The cytokine profiles of conditioned media from mammosphere cultures were determined with cytokine-specific antibodies and cytokine beads. Sulconazole reduced extracellular IL-8 levels in the mammosphere cultures.

**Figure 7 cells-08-01007-f007:**
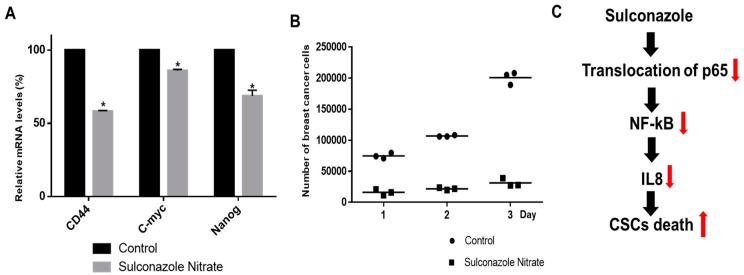
Effects of CSC loads on breast cancer. (**A**) The transcriptional levels of Nanog, C-myc, and CD44 were assayed in sulconazole- and 0.1% DMSO-treated mammospheres using specific primers. β-actin acted as an internal control. The data are presented as the mean ± SD; *n* = 3 independent experiments;* *p* < 0.05 vs. the control. (**B**) Sulconazole prevented mammosphere growth. Sulconazole-treated mammospheres were dissociated into single cells and plated in 6-well plates with equal numbers of cells. The cells were counted in triplicate 1, 2, and 3 days after plating and the mean value was plotted. The data shown represent the mean ± SD of three independent experiments. (**C**) The proposed model for CSC death induced by sulconazole is shown.

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
