# Peer review of "Disruption of the NF-κB/IL-8 Signaling Axis by Sulconazole Inhibits Human Breast Cancer Stem Cell Formation"

_cells, 2019, doi:10.3390/cells8091007_

Round 1

Reviewer 1 Report

This manuscript describes an imidazole-class antifungal drug that has NF-kB inhibitory activity and prevents stem cell growth in breast cancer. The authors carried out a thorough study that included effects of the drug, sulconazole, on cancer cell proliferation and colony formation, migration in culture, apoptosis, tumour growth in nod-skid mice, and mammosphere-forming ability. They also looked at the effect of the drug on stem cell populations. They proposed  a mechanism whereby sulconazole prevents the nuclear translocation of p65 and NF-kB which leads to a reduced IL8, a cytokine survival factor needed for cancer stem cells, its absence causing stem cell death. Flow cytometry was used to show that CD44+CD24 stem cells and ALDH+ stem cells were depleted after addition of the drug. Their results suggest that sulconazole could be an effective therapeutic to block the growth and spread of breast cancer cells.     

Major Comments.

Although DMSO is used as a control for sulconazole, it should be directly stated that the drug was solubilised in DMSO, and that DMSO controls matched the DMSO concentrations used for sulconazole administration. The imidazole-class drug BMS-345541 was referred to in the introduction as an NF-kB inhibitor in lung cancer cells that reduced their stem cell concentrations and self-renewal capacity. Was this inhibitory target of BMS-34554, published in 2018 by Zakaria et al, what originally suggested to the authors that another imidazole drug such as sulconazole might also inhibit NF-kB? It would have been nice to compare the activity of sulconazole to BMS-34554. BMS-34554 was used by Zakaria et al. at a maximum of 10 mM, but appeared to inhibit tumourspheres in lung cancer cells similarly to sulconazole’s effect on mammospheres in breast cancer cells. Fig. 1C, D, E, F, and G do not indicate whether the cells used were MCF-7 or MDA-MB-231B. Presumably, they were not MCF-7 cells in 1F as this cell line is only poorly metastatic. Also, in Fig. 1E and F, the concentration of sulconazole was not stated. In Fig. 2, the in vivo data in mice are presented. Although the dose of sulconazole is given as 10 mg/kg in the Materials and Methods, the dose should also be stated in the legend, and the administration regimen given i.e. one injection per week starting from day 1, as was used in reference 21 (Kim et al. 2019)? In Fig. 3, no details are given on mammosphere culture or the composition of the CSC medium. Also, the MFE calculation should be described. The description of the procedure used for carrying out the flow cytometry in Fig. 4 is minimal, and no concentration is given for the DEAB inhibitor of ALDH. Also, the percentage cells in the four quadrants are very difficult to read. In Fig. 5A and B, it would help to include a histogram of the WB data to show the biological variation in the quantities. Also, in Fig. 1C, CAPE, an NF-kB inhibitor like sulconazole, had no effect on cell proliferation at up to 60 mM, yet even 20 mM CAPE completely knocked out mammosphere formation. Neither changes to Stat-3 nor IL-6 (mRNA and protein levels) were involved in the action of sulconazole according to the results of Fig. 6, although IL-8 was decreased, indicating that CXCR1/IL-8 pathway was being targeted by sulconazole and not the IL-6/IL-6 receptor interaction. Are the CXCR1/IL-8 interaction and IL-6/IL-6 receptor interaction additive? The time axis on Fig. 7B in days should be added to the graph.

Minor Comments.

Although it notes in the methods under statistical analysis that all data are from three independent experiments, the “n = 3 independent experiments” should also be included in the figure legends where appropriate.

Typographical  and grammatical errors.

Line 14    Cancer stem cells (CSCs) are …

L14          expression of another CSC marker, aldehyde dehydrogenase (ALDH) and other

L18          formation, reduced the

L19          NF-kB, and reduced extracellular IL-8 levels in mammospheres. Knocking down

L20          formation and secreted IL-8 levels in mammospheres. These new findings

L29          because of cancer stem cells (CSCs), a

L32          Markers of breast cancer stem cells (BCSCs)

L33          Reduce the font size of “CD44 expression is”

L47          than in normal breast tissue cells,

L161        derived from breast cancer cells (MCF-7 and MDA-MB-231B)

L162        mammospheres declined by 90%.

L250        The cells were counted in triplicate 1, 2, and 3 days after plating.

L266        signaling pathway has been observed

L274        (PDTC), another NF-kB pathway inhibitor, is known to inhibit CSC

Author Response

Point-to-point responses to Reviewers’ comments

Reviewers' Comments:

Review1

This manuscript describes an imidazole-class antifungal drug that has NF-kB inhibitory activity and prevents stem cell growth in breast cancer. The authors carried out a thorough study that included effects of the drug, sulconazole, on cancer cell proliferation and colony formation, migration in culture, apoptosis, tumour growth in nod-skid mice, and mammosphere-forming ability. They also looked at the effect of the drug on stem cell populations. They proposed a mechanism whereby sulconazole prevents the nuclear translocation of p65 and NF-kB which leads to a reduced IL8, a cytokine survival factor needed for cancer stem cells, its absence causing stem cell death. Flow cytometry was used to show that CD44+CD24 stem cells and ALDH+ stem cells were depleted after addition of the drug. Their results suggest that sulconazole could be an effective therapeutic to block the growth and spread of breast cancer cells.     

Major Comments.

 Although DMSO is used as a control for sulconazole, it should be directly stated that the drug was solubilised in DMSO, and that DMSO controls matched the DMSO concentrations used for sulconazole administration.

→The solubility of sulconazole nitrate is 25mg/ml in DMSO. We used sulconazole stock (20mM=9.2mg/ml) and working solutions are 10 μM (0.05 %), 20 μM (0.1%), and 60 μM (0.3%). Solubility of sulconazole nitrate in DMSO is no problem because solubility of sulconazole is 25mg/ml in DMSO. We add DMSO concentrations (20 μM (0.1%) or 60 μM (0.3%) of sulconazole and 0.1 % or 0.3% DMSO at Figure legends.

The imidazole-class drug BMS-345541 was referred to in the introduction as an NF-kB inhibitor in lung cancer cells that reduced their stem cell concentrations and self-renewal capacity. Was this inhibitory target of BMS-34554, published in 2018 by Zakaria et al, what originally suggested to the authors that another imidazole drug such as sulconazole might also inhibit NF-kB? It would have been nice to compare the activity of sulconazole to BMS-34554. BMS-34554 was used by Zakaria et al. at a maximum of 10 mM, but appeared to inhibit tumourspheres in lung cancer cells similarly to sulconazole’s effect on mammospheres in breast cancer cells.

Reviewer comment is reasonable. The imidazole-class drug BMS-345541 may be inhibit breast CSC based on IKK inhibition.

Our paper (Screening of breast cancer stem cell inhibitors using a protein kinase inhibitor library, Cancer Cell Int. 2017 Feb 13;17:25, Choi HS, et al) showed that IKK inhibitor, BAY 11-7082 inhibited breast CSC formation at Figure 2 of our published Cancer Cell Int journal.

The IKK inhibitor approach (BMS-345541, BAY 11-7082, and Sulconazole) will be designed for inhibition assay as a further study and we described the imidazole-class drug BMS-345541 as a potential cancer chemo-preventive agent at discussion section as followed,

The imidazole-class drug BMS-345541, IKK inhibitor as an NF-kB inhibitor reduced their stem cell concentrations and self-renewal capacity in lung cancer cells

Fig. 1C, D, E, F, and G do not indicate whether the cells used were MCF-7 or MDA-MB-231B. Presumably, they were not MCF-7 cells in 1F as this cell line is only poorly metastatic. Also, in Fig. 1E and F, the concentration of sulconazole was not stated.

→ We added MDA-MB-231 at Fig. 1C, D, E, F, and G and the concentration of sulconazole at Fig. 1E and F was added as 20 μM.

In Fig. 2, the in vivo data in mice are presented. Although the dose of sulconazole is given as 10 mg/kg in the Materials and Methods, the dose should also be stated in the legend, and the administration regimen given i.e. one injection per week starting from day 1, as was used in reference 21 (Kim et al. 2019)?

→The reviewer’s point is well taken. We added paragraph used in reference 21 (Kim et al. 2019) at the Figure 2 legend as followed,

The dose of drug used was 10 mg/kg once a week.

In Fig. 3, no details are given on mammosphere culture or the composition of the CSC medium. Also, the MFE calculation should be described.

→The reviewer’s point is well taken. We added paragraph containing mammosphere culture, the composition of the CSC medium and the MFE calculation at the Figure 3 legend as followed,

To establish mammospheres, MCF-7 and MDA-MB-231 cells were seeded at a density of 4x104 and 1x104 cells/well, respectively, in ultralow attachment 6-well plates containing 2ml of complete MammoCultTM medium (StemCell Technologies) which was supplemented with 4 μg/ml heparin, 0.48 μg/ml hydrocortisone, 100 U/ml penicillin, and 100 μg/ml streptomycin. The cells were incubated for 7 days in a 5% CO2 incubator at 37°C.

(MFE, % of control), which corresponds to the number of mammospheres per well/the number of total cells plated per well x100 as previously described [22].

The description of the procedure used for carrying out the flow cytometry in Fig. 4 is minimal, and no concentration is given for the DEAB inhibitor of ALDH. Also, the percentage cells in the four quadrants are very difficult to read.

→The reviewer’s point is well taken. We added paragraph containing the description of the procedure used for carrying out the flow cytometry and the DEAB concentration at the Figure 4 legend. We added a clear flow cytometry data

In Fig. 5A and B, it would help to include a histogram of the WB data to show the biological variation in the quantities.

→We added a histogram of the WB data at Fig.5 A and B

Also, in Fig. 1C, CAPE, an NF-kB inhibitor like sulconazole, had no effect on cell proliferation at up to 60 mM, yet even 20 mM CAPE completely knocked out mammosphere formation.

→ Reviewer comment is reasonable. In order to select more specific CSCs inhibitor without affecting the growth of differentiated cancer cell, we screened many inhibitors containing sulconazole. Using this method, we selected CSCs inhibitors and published (Screening of breast cancer stem cell inhibitors using a protein kinase inhibitor library, Cancer Cell Int. 2017 Feb 13;17: 25, Choi HS, et al).

Neither changes to Stat-3 nor IL-6 (mRNA and protein levels) were involved in the action of sulconazole according to the results of Fig. 6, although IL-8 was decreased, indicating that CXCR1/IL-8 pathway was being targeted by sulconazole and not the IL-6/IL-6 receptor interaction. Are the CXCR1/IL-8 interaction and IL-6/IL-6 receptor interaction additive? The time axis on Fig. 7B in days should be added to the graph.

 → Reviewer comment is reasonable. We expected that NF-kb downregulation by sulconazole can reduce IL-6 and IL-8, but only IL-8 was reduced.

Our interpretations are

I) Major CXCR1/IL-8 interaction and additive IL-6/IL-6 receptor interaction because extracellular IL-8 concentration is higher than IL-6 level. II) CXCR1/IL-8 signal regulate IL-6/IL-6 receptor signal pathway.

(Regulation of cancer stem cells by cytokine networks: attacking cancer's inflammatory roots. Clin Cancer Res. 2011;17 (19):6125-9)

We need further study for crosstalk of IL-8 and IL-6.

 → We added time axis on Fig.7B graph.

Although it notes in the methods under statistical analysis that all data are from three independent experiments, the “n = 3 independent experiments” should also be included in the figure legends where appropriate.

 →We added the n = 3 independent experiments at Figure legends where appropriate.

.

Typographical and grammatical errors.

 Line 14    Cancer stem cells (CSCs) are …

L14          expression of another CSC marker, aldehyde dehydrogenase (ALDH) and other

L18          formation, reduced the

L19          NF-kB, and reduced extracellular IL-8 levels in mammospheres. Knocking down

L20          formation and secreted IL-8 levels in mammospheres. These new findings

L29          because of cancer stem cells (CSCs), a

L32          Markers of breast cancer stem cells (BCSCs)

L33          Reduce the font size of “CD44 expression is”

L47          than in normal breast tissue cells,

L161        derived from breast cancer cells (MCF-7 and MDA-MB-231B)

L162        mammospheres declined by 90%.

L250        The cells were counted in triplicate 1, 2, and 3 days after plating.

L266        signaling pathway has been observed

L274        (PDTC), another NF-kB pathway inhibitor, is known to inhibit CSC

 →We changed Typographical and grammatical errors as your suggestion.

Reviewer 2 Report

The manuscript "Disruption of the NF-kB.....stem cell formation" describes the effect of anti fungal agent sulconazole on breast cancer stem cells. Authors show with in vitro and in vivo mouse tumor growth studies that the agent causes tumor cell growth suppression. They demonstrate that the agent inhibits the expression of IL-8 and cancer stem cell markers CD44 and ALDH.

This is a cell line based data and with limited animal study, clinical utility of sulconazole is limited. It is not clear whether the authors are indicating that sulconazole is important for further studies or NF-kB/IL-8 pathway is important for cancer cell growth suppression. If it is NF-kB/IL-8 pathway then numerous studies have indicated the importance of this pathway for tumor cell growth and inhibitors are available to block this pathway.

More details are required in the introduction on the biochemical studies available for sulconazole. This would point to the importance of this agent in cancer cell suppression through multiple pathways or only through the NF-kB/IL-8 signaling.

For the nude mouse studies, the method section says that tumor size was measured after 1.5 months. However, figure 2 shows measurement was made every 3 to 4 days starting from day 10. This should be included in the methods. 

It is not stated whether nude mouse studies represent sub-cutaneous tumor formation

It is not clear how often and in what volume, sulconazole was injected into the mice. The route of injection and the solvent used are not included. It is assumed that the agent was dissolved in DMSO and diluted in saline for injections.  Calculation shows 200ug of the agent was injected for a 20gm mouse and this information should be clearly mentioned in the methods.

It is also not clear how many animals were included in the control and the drug treated groups. Tumor volume error bars indicate that they represent standard errors and not standard deviation. This should be included in the y axis of the figure as Tumor volume (mm3) (+/-S.E). 

Effect on ALDH shown in figure 4B is not convincing

Reduction of p65 expression in figure 5B needs improvement which is also reflected in unconvincing mammoshpere data with the siRNA. Authors could try 2 or more p65 siRNAs for this experiment.

Author Response

Point-to-point responses to Reviewers’ comments

Reviewers' Comments:

Review2

The manuscript "Disruption of the NF-kB.....stem cell formation" describes the effect of anti fungal agent sulconazole on breast cancer stem cells. Authors show with in vitro and in vivo mouse tumor growth studies that the agent causes tumor cell growth suppression. They demonstrate that the agent inhibits the expression of IL-8 and cancer stem cell markers CD44 and ALDH.

This is a cell line based data and with limited animal study, clinical utility of sulconazole is limited. It is not clear whether the authors are indicating that sulconazole is important for further studies or NF-kB/IL-8 pathway is important for cancer cell growth suppression. If it is NF-kB/IL-8 pathway then numerous studies have indicated the importance of this pathway for tumor cell growth and inhibitors are available to block this pathway.

→ The imidazole antifungal drug (clotrimazole and ketoconazole) inhibited cell proliferation and invasion of human breast cancer (Biomol Ther (Seoul). Bae SH, et al; 2018 Sep 1;26(5):494-502).

A triazole, itraconazole, a commonly used antifungal drug that inhibits Hedgehog pathway activity and cancer growth (Cancer Cell.2010:17(4):388-99).

→ Cancer stem cells were regulated by inflammatory cytokine networks. IL-8 to CXCR1/2 activates NF-κB pathway via Stat3 and Akt signaling. IL-6 and IL-8 production by NF-κB generates a positive feedback loop which maintains constitutive pathway activation. These pathways including IL-1, IL-6 and IL-8, which in turn activate Stat3/NF-κB pathways in both tumor and stromal cells, provide potential targets for the development of novel strategies to target the CSC populations (Regulation of Cancer Stem Cells by Cytokine Networks: Attacking Cancers Inflammatory Roots. Clin Cancer Res. 2011 Oct 1; 17(19): 6125–6129)

Therefore, NF-kB/IL-8 pathway is important for cancer cell and CSCs growth.

More details are required in the introduction on the biochemical studies available for sulconazole. This would point to the importance of this agent in cancer cell suppression through multiple pathways or only through the NF-kB/IL-8 signaling.

→ We added biochemical studies of sulconazole and antifungal imidazole drugs for cancer cell suppression at the introduction as followed,

Azole compounds used as antifungal drugs inhibit the ergosterol biosynthesis pathway through suppression of the enzyme lanosterol 14-α-demethylase, a cytochrome P450 (CYP) enzyme. Azole antifungal drugs consist of an imidazole (clotrimazole and ketoconazole) and a triazole (fluconazole and itraconazole). Recently, antifungal imidazole drugs have well-established pharmacokinetic profiles and known toxicity, which can make these generic drugs strong candidates for repositioning as antitumor therapies [15].

For the nude mouse studies, the method section says that tumor size was measured after 1.5 months. However, figure 2 shows measurement was made every 3 to 4 days starting from day 10. This should be included in the methods. It is not stated whether nude mouse studies represent sub-cutaneous tumor formation. It is not clear how often and in what volume, sulconazole was injected into the mice. The route of injection and the solvent used are not included. It is assumed that the agent was dissolved in DMSO and diluted in saline for injections.  Calculation shows 200ug of the agent was injected for a 20gm mouse and this information should be clearly mentioned in the methods.

→The reviewer’s point is well taken. We added paragraph at the methods as followed,

Nude mice (n=6) received sulconazole using mammary fat pad injection with an optimized dosage of 10 mg/kg. The dose of drug used was 10 mg/kg (200 μg/100 μl) once a week. The measurement was made every 3 to 4 days starting from day 10. The solvent used is DMSO.

It is also not clear how many animals were included in the control and the drug treated groups. Tumor volume error bars indicate that they represent standard errors and not standard deviation. This should be included in the y axis of the figure as Tumor volume (mm3) (+/-S.E). 

→The reviewer’s point is well taken. We added new paragraphs at the methods and Figure 2 and Method.

Effect on ALDH shown in figure 4B is not convincing

→DEAB, ALDH inhibitor treatment without sulconazole reduced ALDH positive cell population from 3.2% to 0 %. Our assay system is working.

Sulconazol treatment reduced ALDH positive cell population from 3.2% to 1.5%. We added a box (red dot line box) for clear data presentation at Fig.4 legend.

Suconalzole reduced CSCs population.

Reduction of p65 expression in figure 5B needs improvement which is also reflected in unconvincing mammoshpere data with the siRNA. Authors could try 2 or more p65 siRNAs for this experiment.

→ We bought 3 kinds of p65 siRNAs from company. We transfected siRNAs into breast cancers cell and induced mammosphere formation for 7 day.

Our data showed that mammosphere formation rate was 65% and inhibition rate was 35% under p65 knockdown. Therefore our date is reasonable.

Reviewer 3 Report

MS# cella-578026

Title: Disruption of the KF-kappaB/IL-8 signaling axis by sulconazole inhibits human breast cancer stem cell formation

             This study indicates that sulconazole inhibited the proliferation of breast cancer cells (BCSs) in vitro, the MDA-MB-231 cell-injected tumor growth in vivo, as well as the mammosphere formation from breast cancer stem cells (BCSCs). Moreover, sulconazole led to a drastic reduction in mammosphere growth via targeting the NF-kappaB signaling pathway that linked to the expression of IL-8 but not IL-6. The results are interesting, and thus the manuscript may be acceptable to the journal with minor revisions.

Can the knockdown of IL-8 with a siRNA-based approach affect the growth of mammosphere? Sulconazole had inhibited the cell proliferation of BCSs in vitro and the tumor growth in BCSs-injected nude mice in vivo. These effects are also applicable to the mechanism, NF-kappaB/IL-8 axis as did in mammosphere? More specifically, can the gene knockdown of IL-8 affect the cell proliferation of BCSs?

Author Response

Point-to-point responses to Reviewers’ comments

Reviewers' Comments:

Review3

Title: Disruption of the KF-kappaB/IL-8 signaling axis by sulconazole inhibits human breast cancer stem cell formation

              This study indicates that sulconazole inhibited the proliferation of breast cancer cells (BCSs) in vitro, the MDA-MB-231 cell-injected tumor growth in vivo, as well as the mammosphere formation from breast cancer stem cells (BCSCs). Moreover, sulconazole led to a drastic reduction in mammosphere growth via targeting the NF-kappaB signaling pathway that linked to the expression of IL-8 but not IL-6. The results are interesting, and thus the manuscript may be acceptable to the journal with minor revisions.

Can the knockdown of IL-8 with a siRNA-based approach affect the growth of mammosphere? Sulconazole had inhibited the cell proliferation of BCSs in vitro and the tumor growth in BCSs-injected nude mice in vivo. These effects are also applicable to the mechanism, NF-kappaB/IL-8 axis as did in mammosphere? More specifically, can the gene knockdown of IL-8 affect the cell proliferation of BCSs?

→ IL-8 knockdown and NF-kappaB/IL-8 axis affected the cell proliferation of BCSs at several paper.

I) Regulation of CSCs by inflammatory cytokine networks. IL-8 to CXCR1/2 activates the NF-kB pathway via Stat3 and Akt signaling. IL-8 production by NF-kB generates a positive feedback loop that maintains constitutive pathway activation and drive CSC self-renewal.

(Regulation of Cancer Stem Cells by Cytokine Networks: Attacking Cancers Inflammatory Roots, Clin Cancer Res. 2011 Oct 1; 17(19): 6125–6129).

II) IL-8 knockdown inhibited lung CSC formation (IL-8 regulates the stemness properties of cancer stem cells in the small-cell lung cancer cell line H446, Onco Targets Ther 2018; 11: 5723–5731).

III) Recombinant IL‐8 significantly increased CSC number (IL‐8 and CXCR1 expression is associated with cancer stem cell‐like properties of clear cell renal cancer. J. Pathol. 2019 ;248(3):377-389)

VI) Targeting IL-8 signalling to inhibit breast cancer stem cell activity, Expert Opin Ther Targets. 2013 (11):1235-41 V) Recent advances reveal IL-8 signaling as a potential key to targeting breast cancer stem cells, Breast Cancer Res 2013; 15(4): 210.

Round 2

Reviewer 2 Report

It is recommended that the drug dissolved in DMSO was diluted in 0.9% saline for injection into mice for the tumor suppression. It it was done, it will be good to include the sentence. For siRNA studies, combination of siRNAs could be included to get improved inhibition of gene expression.